# Impact of Androgen Receptor Activity on Prostate-Specific Membrane Antigen Expression in Prostate Cancer Cells

**DOI:** 10.3390/ijms23031046

**Published:** 2022-01-18

**Authors:** Ulrich Sommer, Tiziana Siciliano, Celina Ebersbach, Alicia-Marie K. Beier, Matthias B. Stope, Korinna Jöhrens, Gustavo B. Baretton, Angelika Borkowetz, Christian Thomas, Holger H. H. Erb

**Affiliations:** 1Institute of Pathology, Universitätsklinikum Carl Gustav Carus Dresden, 01307 Dresden, Germany; ulrich.sommer2@uniklinikum-dresden.de (U.S.); Korinna.Joehrens@uniklinikum-dresden.de (K.J.); Gustavo.Baretton@uniklinikum-dresden.de (G.B.B.); 2Department of Urology, Technische Universität Dresden, 01307 Dresden, Germany; Tiziana.siciliano@tu-dresden.de (T.S.); Celina.ebersbach@uniklinikum-dresden.de (C.E.); AliciaMarie.Beier@uniklinikum-dresden.de (A.-M.K.B.); angelika.borkowetz@uniklinikum-dresden.de (A.B.); chistian.thomas@uniklinikum-dresden.de (C.T.); 3Mildred Scheel Early Career Center, Department of Urology, Medical Faculty and University Hospital Carl Gustav Carus, Technische Universität Dresden, 01307 Dresden, Germany; 4Department of Gynecology and Gynecological Oncology, University Hospital Bonn, 53127 Bonn, Germany; Matthias.Stope@ukbonn.de; 5UroFors Consortium (Natural Scientists in Urological Research), German Society of Urology, 14163 Berlin, Germany; 6National Center for Tumor Diseases Partner Site Dresden and German Cancer Center, 69120 Heidelberg, Germany; 7Tumor and Normal Tissue Bank of the University Cancer Center (UCC), University Hospital and Faculty of Medicine, Technische Universität Dresden, 01069 Dresden, Germany

**Keywords:** *FOLH1*, AR, PCa, antiandrogen, androgen deprivation therapy

## Abstract

Prostate-specific membrane antigen (PSMA) is an essential molecular regulator of prostate cancer (PCa) progression coded by the *FOLH1* gene. The PSMA protein has become an important factor in metastatic PCa diagnosis and radioligand therapy. However, low PSMA expression is suggested to be a resistance mechanism to PSMA-based imaging and therapy. Clinical studies revealed that androgen receptor (AR) inhibition increases PSMA expression. The mechanism has not yet been elucidated. Therefore, this study investigated the effect of activation and inhibition of androgen signaling on PSMA expression levels in vitro and compared these findings with PSMA levels in PCa patients receiving systemic therapy. To this end, LAPC4, LNCaP, and C4-2 PCa cells were treated with various concentrations of the synthetic androgen R1881 and antiandrogens. Changes in *FOLH1* mRNA were determined using qPCR. Open access databases were used for ChIP-Seq and tissue expression analysis. Changes in PSMA protein were determined using western blot. For PSMA staining in patients’ specimens, immunohistochemistry (IHC) was performed. Results revealed that treatment with the synthetic androgen R1881 led to decreased *FOLH1* mRNA and PSMA protein. This effect was partially reversed by antiandrogen treatment. However, AR ChIP-Seq analysis revealed no canonical AR binding sites in the regulatory elements of the *FOLH1* gene. IHC analysis indicated that androgen deprivation only resulted in increased PSMA expression in patients with low PSMA levels. The data demonstrate that AR activation and inhibition affects PSMA protein levels via a possible non-canonical mechanism. Moreover, analysis of PCa tissue reveals that low PSMA expression rates may be mandatory to increase PSMA by androgen deprivation.

## 1. Introduction

Prostate cancer (PCa) is the most common cancer in men. According to the “Cancer Today” project, PCa has the second-highest incidence of all cancers in men, with 1,414,259 newly diagnosed cases (Europe 473,344, North America 239,574) and 375,304 deaths yearly related to PCa worldwide (Europe 108,088, USA 37,192) [1]. Local confined PCa is treated by radiotherapy or radical prostatectomy with curative intent [2], whereas locally advanced or metastatic PCa is palliatively treated. The proliferation and progress of PCa are highly dependent on androgens [3]. Therefore, to reduce androgen levels, androgen deprivation therapy (ADT) is the primary treatment option for locally advanced or metastatic PCa [4]. This treatment regimen includes surgical or chemical castration with luteinizing hormone-releasing hormone agonists and antagonists or gonadotropin-releasing hormone agonist and antagonists [2]. Since 2021, it has been recommended to combine ADT with novel hormonal therapy (ADT+NHT) such as the antiandrogens enzalutamide, apalutamide, darolutamide, or the CYP17A1 inhibitor abiraterone [4,5]. In addition, ADT combined with taxan-based chemotherapeutic docetaxel represents a further therapy option. However, ADT alone or in combination inevitably leads to disease progression and the development of castration-resistant PCa (CRPC), which is currently incurable [2,4]. Present day treatment strategies include ADT, enzalutamide, apalutamide, darolutamide, abiraterone, docetaxel, cabazitaxel, olaparib, Radium-223, and Lutetium-177-PSMA [4,6,7].

Prostate-specific membrane antigen (PSMA), also known as Folate Hydrolase 1 or Glutamate carboxypeptidase II, is a transmembrane type II glycoprotein coded by the *FOLH1* gene located in the short arm of chromosome 11 [6,7]. The protein has a unique 3-part structure containing a 19-amino-acid internal portion, a 24-amino-acid transmembrane portion, and a 707-amino-acid external portion [6,7]. It is expressed in various benign and malignant tissues, including, e.g., salivary glands, the duodenal mucosa, proximal renal tubular cells, and neuroendocrine cells [6,7,8]. However, compared with other tissues, PSMA expression significantly increases in prostate cancer (PCa), where it is localized at the luminal surface of the prostatic ducts and presents a large extracellular ligand-binding domain [6,7,8]. The exact biological function of PSMA in PCa is still not wholly understood; however, its possible role as a carboxypeptidase (folate hydrolase) on poly-g-gluta-mated folates, as a peptidase on the acidic neuropeptide N-acetylaspartyl glutamate, in endocytosis transport function, and invasiveness have all been suggested [6,7].

PSMA became one of the most promising targets for imaging and novel therapies in PCa [6,7]. Several studies have revealed that PSMA-targeted imaging has a higher positive predictive value and sensitivity compared to conventional imaging such as multiparametric magnetic resonance imaging (MRI) or computed tomography (CT) [6,7]. In particular, positron emission tomography (PET) fluorine-18 (^18^F)- and ^68^Gallium (^68^Ga)-labelled radiotracers targeting PSMA have demonstrated significantly higher sensitivity. They became part of the primary diagnostic in PCa guidelines for cases of PSA-recurrence after local therapy [2,4]. Besides imaging, PSMA therapy has emerged as a game-changing therapeutic target, where small-molecule inhibitors of PSMA are linked to radioligands [6]. An example of this is Lutetium-177 (^177^Lu), labelled PSMA in metastatic castration-resistant PCa (mCRPC) that delivers β radiation to PSMA expressing cells. Cellular uptake of ^177^Lu labelled PSMA led to a higher PSA response and fewer adverse events than cabazitaxel [9]. Moreover, the VISION trial demonstrated that the addition of ^177^Lu -PSMA-617 to standard care significantly extended survival among patients with mCRPC [10].

In vivo and in vitro studies suggested an influence of the androgen receptor (AR) on PSMA expression [11,12]. The AR belongs to the nuclear receptor family and is activated by androgens. AR activity is controlled by androgen binding followed by nuclear translocation, DNA-binding, and activation of AR-target gene expression [13]. DNA binding of the AR is directed by a 15-bp palindromic sequence, the so-called AR-binding sites (ARBs). ARBs are located in intronic or intergenic regions and contain two hexameric 5′-AGAACA-3′ half-sites with a three base pairs spacer [14,15]. Next to the idealized or generalized (canonical) AR pathway, non-canonical AR pathways have been described. These include signaling crosstalk with the SRC kinase, PI3K Pathway, and ligand-independent AR activation [16]. Progress and growth of PCa are highly dependent on the AR, and therefore representing the main therapeutic target in metastatic PCa [4]. AR-based therapy includes ADT and treatment with antiandrogens. For ADT and enzalutamide, there is evidence that treatment leads to a change in PSMA levels [11]. A high number of PCa patients receiving PSMA-ligand PET or PSMA-guided radiotherapy are on ADT or antiandrogen treatment. The connection between PSMA levels and the AR needs to be clarified [11,17,18].

The study aimed to investigate the influence of androgen activity manipulation by ADT and the antiandrogens bicalutamide, enzalutamide, apalutamide, or darolutamide on PSMA levels in vitro and to compare these findings with PSMA expression in patients receiving systemic therapy. To substantiate our investigation, three AR-positive cell lines (LNCaP, C4-2, LAPC4) and one AR negative cell line (PC3) have been used (Table 1). The LNCaP cell line represents an androgen-sensitive human PCa cell derived from a supraclavicular lymph node metastasis from a 50-year-old Caucasian male in 1977 [19,20]. LNCaP cells express an endogenous mutated AR (T877A), normal prostate epithelial cell markers, and represent a luminal phenotype [20,21,22,23]. The AR-positive cell line C4-2 is a PCa sub-cell line derived from LNCaP cells that grows without androgens but responds to androgen levels [24]. The cell line represents a CRPC status and also has a luminal phenotype [23,24]. Moreover, the C4-2 cells express the ligand-independent AR splice variant V7 (ARv7) protein [25,26]. The cell line LAPC4 has been harvested from a lymph node metastasis of a male patient with CRPC and subsequently xenografted into SCID mice [27,28]. Explants were later used to establish the immortalized PCa cell line. In contrast to the LNCaP and C4-2 cells, LAPC4 cells express wild type AR, luminal and basal cell markers such as cytokeratin 5 [20]. The PC3 cell line was established in 1979 from a PCa bone metastasis of a 62-year-old Caucasian male [29]. The cell line expresses no AR, STAT3, STAT5, and luminal markers [20,30,31]. In addition, PC3 cells express several basal cell markers and have prostatic small cell carcinoma characteristics [32].

## 2. Results

### 2.1. FOLH1 mRNA and PSMA Protein Levels in the PCa Cell Lines LAPC4, LNCaP, C4-2 and PC3

To analyze the influence of increased and decreased AR activity on *FOLH1* expression, the AR-positive castration sensitive cell lines LAPC4, LNCaP, and the castration-resistant LNCaP sub-cell line C4-2 was used [20]. In addition, the AR negative cell line PC3 was used as a negative control for PSMA and AR [20,29,33]. qPCR analysis (Figure 1A) revealed that all AR-positive cell lines express *FOLH1* mRNA at which LAPC4 expressed the lowest and C4-2 the highest amount of *FOHL1* mRNA. On protein level (Figure 1B,C), in contrast, LAPC4 expressed no detectable level of PSMA protein. However, in line with the mRNA data, C4-2 cells expressed the highest level of PSMA protein.

### 2.2. Changes in AR Activity Modulate FOLH1 mRNA Levels

Several studies reported that modulated AR activity could change *FOLH1* transcription [12,34,35,36,37]. Therefore, the cell lines LAPC4, LNCaP, and C4-2 were starved for 24 h with a medium containing 5% FBSdcc and subsequently treated with different concentrations of the synthetic androgen R1881 for 16 h (Figure 2A–C). All tested cell lines showed a concentration-dependent decrease in *FOLH1* mRNA peaking at 1 nM R1881. To investigate if this effect can be rescued by inhibition of the AR, the cells were treated with R1881 combined with the antiandrogens bicalutamide, enzalutamide, apalutamide, or darolutamide. To this end, the cell lines LAPC4, LNCaP, and C4-2 were again starved for 24 h with a medium containing 5% FBSdcc and subsequently treated for 16 h with the synthetic androgen R1881 or with a combination of R1881 and antiandrogens (Figure 2D–F). In LAPC4 (Figure 2D), all tested antiandrogens reversed the inhibitory effect of the R1881 treatment. However, antiandrogen treatment in LNCaP only revealed a significant rescue of the R1881 effects on the *FOLH1* transcription after apalutamide treatment (Figure 2E). Even if not significant, enzalutamide also showed the tendency to reverse the inhibitory effect of the R1881 treatment, whereas bicalutamide and darolutamide showed only minor effects. In C4-2 cells (Figure 2F), except for enzalutamide, all antiandrogens partially abolished the inhibitory effect of R1881 on *FOLH1* transcription.

### 2.3. The AR Is Not Mandatory for FOLH1 Gene Expression

To assess whether the AR can alter the PSMA expression, publicly available AR chromatin immunoprecipitation sequencing (ChIP-Seq) datasets (GSE62442, GSE65066, Figure 3A,B) were analyzed for the loci of *FOLH1* in the AR-positive cell lines LNCaP, C4-2, and 22Rv1 [38]. In the genomic region surrounding *FOLH1*, two AR binding sites (ARBs) have been found in all examined cell lines (Figure 3A), and four AR binding sites are found uniquely in 22Rv1. Analysis of the dataset GSE62442 (Figure 3B) revealed that only at the ARB around exon 11, identified in 22Rv1, was there a peak increase after treatment with 1 nM of the synthetic androgen Mibolerone in the cell lines LNCaP and C4-2. In addition, the GSE62442 dataset also identified three additional ARBs in the proximity of exon 5 and exon 6. Furthermore, AR ChiP-Seq analysis of samples obtained from hormone-sensitive PCa (HSPC) and castration-resistant PCa (CRPC) identified no new ARB (Figure 3C). Motif analysis of the gene sequence revealed that the hexameric 5′-AGAACA-3′ half-sites are distributed over the whole *FOLH1* gene (Appendix A). However, the canonical ARB was not found.

To assess if the expression of the AR is mandatory for *FOLH1* mRNA expression, AR knockdown and overexpression were investigated. To this end, the datasets GSE11428 and GSE13332 were analyzed for changes in *FOLH1* mRNA levels. These datasets reveal no change in *FOLH1* mRNA levels after AR knockdown (Figure 4A). Analysis of the dataset GSE15091 also indicated no change in *FOLH1* mRNA levels after overexpression of the AR in PC3 (Figure 4B). As knockdown and overexpression did not change *FOLH1* mRNA levels, AR and *FOLH1* mRNA levels were investigated in different tissue using the Affymetrix tissue dataset published on the publicly available database Project Betastasis (https://www.betastasis.com/tissues/affymetrix_tissue_dataset/, accessed on 24 October 2021). No correlation could be shown between AR and *FOLH1* mRNA expression (Figure 4C,D).

### 2.4. Impact of AR Activity on PSMA Protein Levels

As the inhibitory effect of R1881 on *FOLH1* mRNA levels could at least partially be rescued by antiandrogens, the influence of R1881 and the combination of R1881 with antiandrogens on protein levels were assessed using western blot. In line with the results of the cell line comparison (Figure 1B,C), no PSMA protein could be detected (Figure 5A) after 48 h androgen deprivation. In LNCaP cells, 48 h treatment with 1 nM R1881 resulted in a significant decrease in PSMA protein levels (Figure 5B,D). This effect could be reduced to a small extent by treatment with antiandrogens (Figure 5D). Similar effects could be observed in cell line C4-2. Following 48 h of treatment with 1 nM R1881 PSMA protein levels significantly decreased (Figure 5C,E). Identical to the LNCaP cells, antiandrogen treatment abolished the effects of 1 nM R1881 in C4-2 cells partially.

### 2.5. Inhibition of AR Activity Does Not Generally Change PSMA Protein Levels in PCa Tissue

Our western blot results (Figure 5) revealed a change in PSMA protein expression after activation and inhibition of the AR in LNCaP and C4-2. PSMA protein was assessed in PCa patients’ tissue specimens before and during androgen deprivation therapy (Figure 6A,B). The PCa cohort included 34 treatment-naive samples, 55 tissue samples of patients under ADT, and 15 tissue samples from patients under ADT+NHT. Therefore, the patients were still under treatment as they received transurethral resection of the prostate (TURP). In the used cohort, ADT was induced by treatment with buserelin, triptorelin, degarelix or leuprorelin. For ADT+NHT, ADT combined with enzalutamide or abiraterone was administered. The treatment regimen was used for at least 1 month. To assess the influence of androgen manipulation on PSMA protein expression, samples from treatment-naive HSPC and ADT/ADT+NHT treated PCa samples (*n* = 70) were analyzed (Figure 6C). The data demonstrated an insignificant and marginal increase in the IRS from a mean of 9.97 ± 3.25 to 10.10 ± 3.50. To assess if further treatment with antiandrogens impacts PSMA expression, the ADT/ADT+NHT cohort was subgrouped into ADT and ADT+NHT. The treatment duration was over at least a one month period. In line with the preceding result, ADT+NHT marginal and insignificant increased the PSMA protein expression from an IRS 10.07 ± 3.79 to 10.20 ± 2.21 (Figure 6D). As a general comparison of the cohorts only revealed a slight difference in PSMA protein expression, TURP tissue of 6 patients before and under ADT was assessed (Figure 6E). PSMA protein expression was elevated after ADT treatment in 3 out of 6 patients (Figure 6A). These 3 patients showed a mean IRS of 7.67 ± 1.53, whereas the patients with no change in PSMA had an IRS of 12.00 ± 0.00. Vice versa, only one patient (IRS 12) showed a decrease in PSMA protein expression after ADT. Two patients showed no change in PSMA levels (Figure 6B).

## 3. Discussion

PSMA has become an important target in PCa diagnostics and therapy [6,7]. Both diagnostic and therapeutic approaches take advantage of the high PSMA protein expression on PCa cells and, therefore, small metastasis can be targeted. However, in line with reported PCa heterogeneity, PSMA is also heterogeneously expressed. Low PSMA protein levels have been reported as possible resistance mechanisms to PSMA-based radioligand diagnostic and therapies [39,40,41,42]. For this reason, the study of *FOLH1* gene regulation came into scientific focus to improve the effectiveness of PSMA-targeted diagnostics and therapies. This work also focused on regulating the *FOLH1* gene and how its expression can be affected by manipulating AR-mediated gene transactivity with androgen depletion and the antiandrogens bicalutamide, enzalutamide, apalutamide, and darolutamide. Regulation of the PSMA protein by androgen manipulation has been previously reported, and therefore antiandrogen therapy has been discussed as a possible enhancer for PSMA-based theragnostic [6,7,11,36]. For example, enzalutamide has been reported to enhance the PSMA protein expression of PCa cells with low PSMA protein levels. Therefore, enzalutamide treatment may be a way to overcome resistance to PSMA-targeted diagnostics and therapies [11]. Two castration-sensitive PCa cell lines with non-detectable PSMA protein level (LAPC4) and average PSMA protein level (LNCaP) were used to investigate the effects of manipulating the AR activity. In addition, the LNCaP sub-cell line C4-2 was used, representing a castration-resistant cell model with high PSMA protein expression. In line with the data of Evans et al., treatment with the synthetic antiandrogen R1881 downregulated *FOLH1* mRNA in all tested cell lines [43]. This effect could only entirely be rescued in LAPC4 cells by all used antiandrogens. In LNCaP cells, only enzalutamide and apalutamide could partially reverse the effect of R1881 on *FOLH1* mRNA levels, whereas, in C4-2, only enzalutamide showed no impact. Cell line-specific effects of antiandrogens have been identified in a previous study and explained with the AR mutation status and AR protein stability after antiandrogen treatment [44]. AR point mutations have been reported to change transcriptional activity, ligand specificity, and response to antiandrogens [45]. For example, the F877L mutation alters the antagonistic effects of the antiandrogens to agonistic effects of the antiandrogen enzalutamide and apalutamide [45,46,47,48]. LNCaP and C4-2 cells express the AR T877A mutant, which has been associated with low antiandrogen response [45,48]. This mutation may partially explain the different effects of antiandrogens on FOHL1 in LNCaP and C4-2. Moreover, C4-2 cells express the ligand-independent AR splice variant V7 protein reported to change full-length AR activity and antiandrogens response [25,26].

These qPCR results demonstrated regulation of *FOLH1* mRNA levels in the tested cell lines independent of castration status. However, not all cell lines responded to the tested antiandrogens; therefore, general antiandrogen control of *FOLH1* mRNA levels cannot be assumed.

As the AR is targeted by R1881 and antiandrogens directly, two publicly available ChiP-Seq (GSE62442, GSE65066, GSE28219) databases were used to analyze the *FOLH1* locus for ARBs. The AR ChIP-Seq analysis identified several ARBs among multiple areas of the *FOLH1* gene in PCa cell lines. However, none of these ARBs are located in the *FOLH1* promoter or *FOLH1* enhancer, located within the third intron of *FOLH1* [11,49,50]. However, the previous analysis of these regulatory sequences revealed a half-site ARB in the enhancer element, which could not be identified by the analyzed ChiP-Seq data [51]. Therefore, it is hypothesized that *FOLH1* gene transcription is either regulated by untypical ARBs or through competition with other regulatory proteins such as AP1 and CREB [51,52,53,54]. To underpin the ChIP-Seq results, we analyzed AR-positive PCa cell lines after AR-knockdown and AR-negative PCa cell lines after AR overexpression. Neither the knockdown nor overexpression experiments changed *FOLH1* mRNA levels. Interestingly, estrogen treatment of the PSMA and AR negative cell line PC3 increased the PSMA protein expression in these cells, indicating that the *FOLH1* gene does not harbor a deletion mutation nor is it epigenetically silenced [33]. Together with the fact that ChiP-Seq analysis should also reveal untypical ARBs, AR and *FOLH1* mRNA expression do not correlate in different tissue, and modulation of AR expression did not influence *FOLH1* mRNA expression, there is strong evidence that the *FOLH1* gene is regulated through competition of the AR with other regulatory proteins. In addition, inhibition of vascular endothelial growth factor receptor tyrosine kinases by cediranib has also been reported to regulate *FOLH1* in Alveolar soft part sarcoma [55]. As AR also regulates several kinases such as SRC and PI3K, there is also the possibility that the observed effects on *FOLH1* mRNA are due to the non-genomic AR pathway. However, more investigations are needed to strengthen this hypothesis.

Furthermore, as the mechanism is not fully solved yet, the already published data has revealed that *FOLH1* mRNA is regulated by modulation of the AR activity in in vitro models [11,43,49,50]. In contrast to in vivo and patient data, the low PSMA protein expression of LAPC4 could not be increased to a detectable level by androgen deprivation or antiandrogen treatment [36]. Low PSMA protein in a small cell subpopulation has been demonstrated previously in the cell lines LAPC4 and 22Rv1 [56]. Phenotype analysis of LAPC4 cells have demonstrated that LAPC4 expresses luminal and basal cell markers; therefore, the low PSMA protein expression could be a consequence of partial dedifferentiation or a stem cell-like transition [20]. This hypothesis is strengthened by the observations that basal epithelium and stromal cells are PSMA negative [57]. In contrast, androgen treatment reduced PSMA protein expression, as shown by Evans et al., an effect which could be partially rescued by treatment with an antiandrogen. However, there was no general rescue by all antiandrogens on protein level, and each antiandrogen showed different effects in the used cell lines. Moreover, the impact of the antiandrogens on the protein was not in line with the mRNA data presented in this study. The discrepancy may be due to the different incubation times for the mRNA and protein experiments and the cellular background of the AR-positive cell lines.

The data in this study could not generalize the regulating effects of antiandrogens on PSMA protein levels. However, direct regulation of PSMA by the synthetic androgen R1881 could be shown in the cell lines LNCaP and C4-2. Therefore, tissue specimens of treatment-naive and ADT PCa patients were analyzed for PSMA levels. Interestingly, the suggested increase in PSMA level after ADT or ADT+NHT could not be confirmed here [11,35,36,58]. However, closer tissue inspection from the same patients with a low PSMA IRS before and during ADT reveals an increase in PSMA expression after ADT. In contrast, patients with a high PSMA IRS had a stable or a decreased PSMA expression after ADT. This heterogenous regulation of PSMA by ADT have been reported before. In line with the results here, Staniszewska and colleagues reported an increase in PSMA levels after blockade of AR signaling in patients with low baseline tumoral PSMA levels. In contrast, several reports have shown a decrease in PSMA levels after ADT [59,60]. However, due to the small cohort size, no generalization is possible if AR-mediated regulation of PSMA is dependent on the PSMA level. Therefore, further investigations on a larger patient cohort are necessary to reveal if PSMA levels may predict changes of PSMA after manipulating the AR.

In this study, the influence of the manipulation of the AR activity on the expression of *FOLH1* in vivo and patients’ specimens was assessed. The data demonstrated that activation and inhibition of the AR influence PSMA protein levels by a possible non-canonical mechanism. Moreover, analysis of PCa tissue reveals that low PSMA expression may be mandatory to increase PSMA by ADT. However, more studies are required to decipher how the AR influences PSMA to recommend ADT before diagnostics or treatment with PSMA-ligands. Moreover, as the endocytosis transport function is mandatory for PSMA ligand-based methods, the relationship between AR signaling and the transport function of PSMA needs further investigation.

## 4. Materials and Methods

### 4.1. Cell Culture

The human PCa cell lines LNCaP and PC3 were obtained from the American Type Culture Collection. C4-2 cells were kindly provided by Prof. Thalmann (University of Berne, Switzerland) [24]. LAPC4 cells were provided by Dr A. Cato (University of Karlsruhe, Karlsruhe, Germany). LNCaP and C4-2 have been chosen as they represent a connected cell model system representing the castration sensitive (LNCaP) and castration-resistant (C4-2) PCa. Moreover, C4-2 express ARv7. LAPC4 has been selected as it expresses a wild type AR. PC3 has been chosen as AR and PSMA negative control. LAPC4, LNCaP, C4-2, and PC3 were cultured as described in Erb et al. 2020 [30]. Characteristics of the cell lines have been previously described in Table 1. All cell lines were maintained at 37 °C in 5% CO_2_. Mycoplasma testing was performed using the Mycoalert Detection Assay (Lonza, Basel, Switzerland). STR profiling was used to verify cell line authentication.

### 4.2. ChIP-Seq Analysis and Data Mining

For AR ChIP-Seq analysis, the AR binding sites were extracted from the publicly available GEO dataset (GSE62442, GSE65066, GSE28219) and the data were visualized with the Integrative Genomics Viewer [38,61,62].

For PSMA expression analysis, the public datasets GSE11428, GSE13332, and GSE15091 have been used. GSE11428 investigated AR-induced gene expression in PCa cells LNCaP and LNCaP abl by transfecting siRNA against AR [63]. GSE13332 explores gene expression in C4-2 cells after AR knockdown [64]. GSE15091 analyzed PC3 cells harboring the wild type-AR construct in the growth conditions of 10nM R1881 and ethanol (the solvent for R1881) [65].

### 4.3. Drug Treatment

R1881 (Sigma-Aldrich, St. Louis, MO, USA, R0908-10MG, Lot No: 085M4610V), bicalutamide (Selleck Chemicals LLC., Houston, TX, USA, S1190, Lot Number: 5), enzalutamide (Astellas Pharma, 3343, Lot Number: RS-8BK0189-4), apalutamide (Selleck Chemicals LLC., S2840, Lot Number: 2), and darolutamide (Selleck Chemicals LLC., S75559, Lot Number: 1) were dissolved in DMSO (100 mM stock solution), and aliquots were stored at −80 °C.

Bicalutamide is a first-generation non-steroidal antiandrogen used to treat metastatic and local advanced PC combined with LHRH analogues or orchiectomy [66]. While the exact mechanism of action is still elusive, it is suggested that bicalutamide stimulates the AR to bind DNA and mediates the recruitment of corepressors (e.g., NCoR) instead of coactivators (e.g., CBP or p300) [67,68].

The second-generation antiandrogen enzalutamide is indicated for metastatic PC [66]. Enzalutamide blocks the ligand-binding domain of the AR, preventing the binding of androgens such as 5α-dihydrotestosterone. In addition, it inhibits nuclear translocation of the AR and, therefore, DNA binding and coactivator recruitment [67,68].

Apalutamide is a second-generation antiandrogen indicated for metastatic PC treatment [69]. Its mechanism of action includes binding to the AR ligand-binding domain, preventing AR activation, inhibiting the AR nuclear translocation, and DNA binding, therefore inhibiting the androgen receptor-mediated transcription [70,71].

Darolutamide is a non-steroidal AR antagonist used in metastatic PC and inhibits nuclear translocation of the AR, thereby decreasing the activation of genes required for the growth and survival of PC cells [72].

The medium was changed to 5% dextran-coated charcoal treated FBS (FBSdcc; Thermo Fisher Scientifc, Waltham, MA, USA) 24 h after seeding cells to deplete the steroid hormones and growth factors. After 24 h, for qPCR, cells were subsequently treated for 16 h with the synthetic androgen 1 nM R1881 or with a combination of 1 nM R1881 and 10 µM antiandrogens. For western blot analysis, cells were subsequently treated for 48 h with the synthetic androgen 1 nM R1881 or with a combination of 1 nM R1881 and 10 µM antiandrogens.

### 4.4. Western Blot Analysis

As described earlier, subcellular fractionation, cell harvest, protein determination, and western blot were performed [30]. SDS-gel separated 20 µg protein lysate electrophorese using NuPAGE™ 4–12% Bis-Tris protein gels and transferred to a nitrocellulose membrane using the iBlot dry blotting system (all Thermo Fisher Scientifc, Waltham, MA, USA). Additionally, 5 µL Spectra Multicolour Broad Range (Thermo Fisher Scientifc, Waltham, MA, USA) protein standard and 1 µL MagicMark™ XP western protein standard (Thermo Fisher Scientifc, Waltham, MA, USA) were used. For detection, the membranes were incubated with WesternBright Sirius HRP substrate (Advansta), and signals were detected by a Microchemi chemiluminescence system (DNR Bio-Imaging Systems, Ha-Satat, Israel). The antibodies used are listed in Table 2. Densitometric analysis of experiments was performed with the Image-Studio Lite 5.2 software (LI-COR, Lincoln, NE, USA). Uncropped western blot images are displayed in the Appendix A. Raw images files are displayed in S1_raw_images.

### 4.5. RNA Isolation and Quantitative Real-Time PCR

Total RNA was isolated using the DIRECT-ZOL RNA MINIPREP (Zymo Research, Freiburg im Breisgau, Germany) following the manufacturer’s instructions. cDNA synthesis was performed with 500 ng total RNA using the Superscript II RNase H Reverse Transcriptase kit (Thermo Fisher Scientifc, Waltham, MA, USA). The quantitative real-time polymerase chain reaction (qPCR) was performed with GoTaq Probe qPCR Master Mix (Promega, Mannheim, Germany) and appropriate primers for 45 cycles on a LightCycler 480 instrument (Roche, Mannheim, Germany). The geometric mean of HPRT1 and TBP was used for normalization. The LightCycler® 480 Software, Version 1.5 (Roche, Mannheim, Germany) was used for the determination of crossing point (Cp) values. ΔCp = Cp_GOI_-Cp_Housekeeper_ values were calculated and expressed as 2^−ΔCp^. The following primer assays have been used (all Thermo Fisher Scientifc, Waltham, MA, USA): *FOLH1* (Hs00379515_m1), *HPRT1* (Hs02800695_m1), and *TBP* (Hs00427620_m1).

### 4.6. Patient Material and Immunohistochemistry (IHC)

Patients’ samples were selected from the Tumor and Normal Tissue Bank of the University Cancer Center Dresden. The Ethics Committee approved the use of archived material of the Medical University of Dresden (Study no. EK59032007, 6 March 2007). Written consent was obtained from all patients and documented in the medical hospital *Carl Gustav Carus* Dresden database according to statutory provisions. The cohort contained 105 tissue specimens of 97 PCa patients undergoing TURP recruited from 2011 and 2020. Tissue blocks were cut in serial sections of 1–2 µm thickness; sections were deparaffinized with BenchMark XT (Ventana Medical Systems, Oro Valley, AZ, USA) and then exposed to a heat-induced epitope retrieval. The PSMA (Clone 3E6; LOT: 11292493) antibody (Dako Agilent, Frankfurt am Main, Germany) was used in a 1:50 dilution for staining, followed by counterstaining with hematoxylin, dehydration, and mounting of the slides. For PSMA, IHC was evaluated by the department of pathology Dresden using the following modified “quick-score” protocol: staining intensity was scored 0–4 (0= absent, 1 = weak, 2 = intermediate, 3 = strong, very strong = 4). The percentage of positively stained cells was scored 0–4 (0 = absent, 1 ≤ 10%, 2 ≤ 50%, 3 ≤ 75%, 4 > 75%). Both scores were multiplied to obtain an immunoreactivity score (IRS), ranging from 0–16.

### 4.7. Statistical Analysis

Prism 9.2 (GraphPad Software, San Diego, CA, USA) was used for statistical analyses. Differences between treatment groups were analyzed using ordinary a one-way ANOVA or Student’s *t*-test. Data are presented as mean ± s.e.m. to estimate the various means in multiple repeated experiments [73]. *p*-values of ≤0.05 were considered statistically significant. All differences highlighted by asterisks were statistically significant as encoded in figure legends (* *p* ≤ 0.05; ** *p* ≤ 0.01; *** *p* ≤ 0.001). All experiments have been performed in at least three biological replicates unless noted otherwise

## Figures and Tables

**Figure 1 ijms-23-01046-f001:**
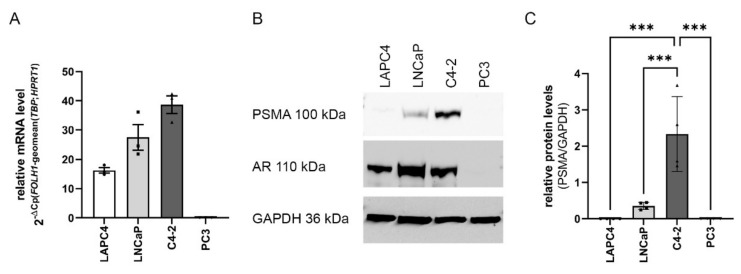
*FOLH1* mRNA and PSMA protein levels in the AR-positive PCa cell lines LAPC4, LNCaP, C4-2, and PC3: (**A**) *FOLH1* mRNA levels (*n* = 3) in LAPC4, LNCaP, C4-2, and PC3, cell lines cultures normalized to the geometric mean of TBP and HPRT1. Values are expressed as mean ± SEM. (**B**) Representative western blot for PSMA, AR, and GAPDH expression in LAPC4, LNCaP, C4-2, and PC3. (**C**) Densiometric analysis of PSMA western blot analysis normalized to GAPDH. Values are expressed as mean ± SD. All differences highlighted by asterisks were statistically significant (*** *p* ≤ 0.001).

**Figure 2 ijms-23-01046-f002:**
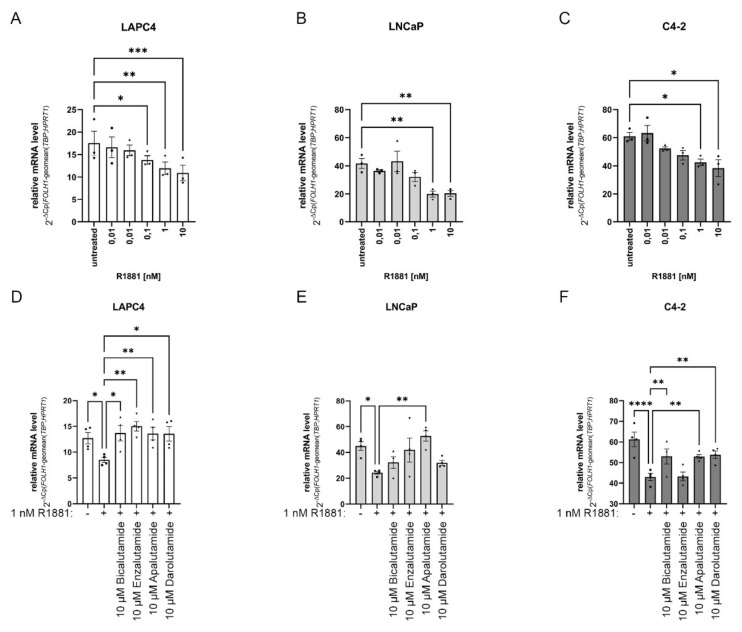
Changes in AR activity modulate *FOLH1* mRNA levels: (**A**–**C**) Change of the *FOLH1* mRNA levels after 16 h treatment with different concentrations of R1881 in the cell lines LAPC4 (**A**), LNCaP (**B**), and C4-2 (**C**). Values are expressed as mean ± SEM of at least three independent experiments. (**D**–**F**) Change of the *FOLH1* mRNA levels after 16 h treatment with different concentrations of 1 nM R1881 and 10 µM bicalutamide, 10 µM enzalutamide, 10 µM apalutamide, or 10 µM darolutamide in the cell lines LAPC4 (**A**), LNCaP (**B**), and C4-2 (**C**). Values are expressed as mean ± SEM of at least three independent experiments. All differences highlighted by asterisks were statistically significant (* *p* ≤ 0.05; ** *p* ≤ 0.01; *** *p* ≤ 0.001; **** *p* ≤ 0.0001).

**Figure 3 ijms-23-01046-f003:**
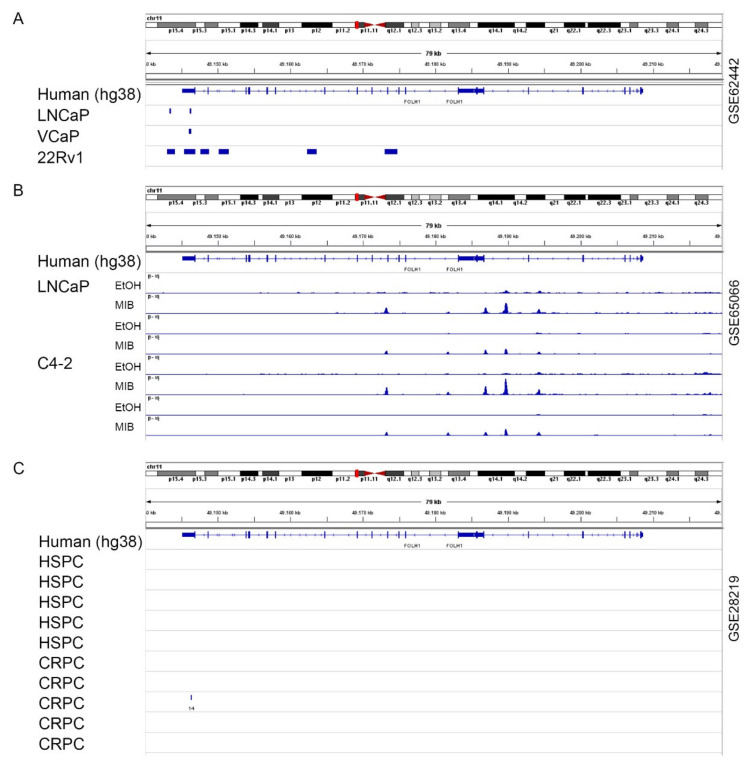
The AR is not mandatory for *FOLH1* gene expression: (**A**) Visualization of the AR binding sites from a publicly available ChIP-Seq dataset (GSE62442) at the *FOLH1* locus of the AR-positive cell lines LNCaP, VCaP, and 22Rv1, which were treated with vehicle or 1 nM R1881 for 1 h. (**B**) Visualization of AR chromatin binding and normalized peak heights from a publicly available ChIP-Seq dataset (GSE65066) at the *FOLH1* locus of the cell lines LNCaP and C4-2, which were treated with vehicle (EtOH) or 1 nM of the synthetic androgen Mibolerone. (**C**) AR ChiP-Seq analysis of samples obtained from hormone-sensitive PCa (HSPC) and castration-resistant PCa (CRPC) taken from the publicly ChIP-Seq dataset GSE28219.

**Figure 4 ijms-23-01046-f004:**
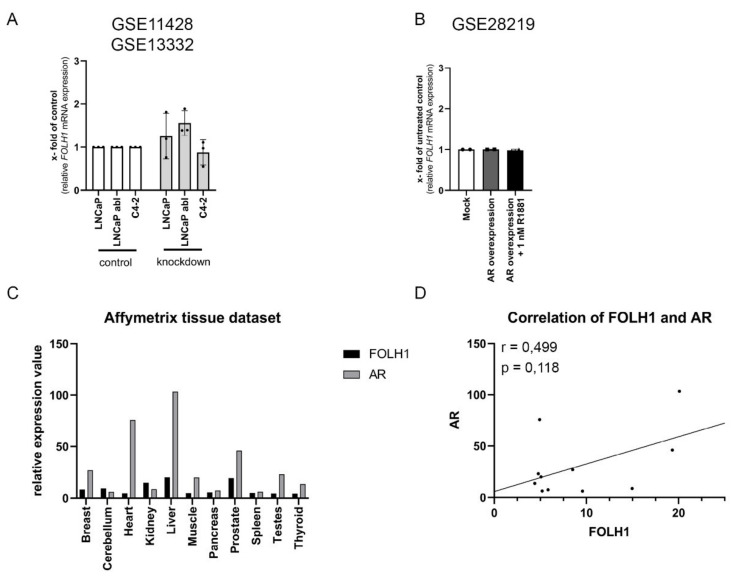
Verification of the necessity of the AR for the expression of the *FOLH1* gene. (**A**) Analysis of *FOLH1* mRNA expression in several AR-positive cell lines after AR-knockdown. Values are expressed as mean ± SD. (**B**) Analysis of *FOLH1* mRNA expression in the AR-negative cell line PC3 after AR overexpression and AR overexpression combined with 1 nM R1881 treatment. (**C**) Analysis of the AR and *FOLH1* mRNA levels in different tissue using the Affymetrix tissue dataset published on the publicly available database Project Betastasis (https://www.betastasis.com/tissues/affymetrix_tissue_dataset/, accessed on 24 October 2021). (**D**) Correlation of AR and FOLH1 mRNA levels in different tissue obtained from the publicly available database Project Betastasis.

**Figure 5 ijms-23-01046-f005:**
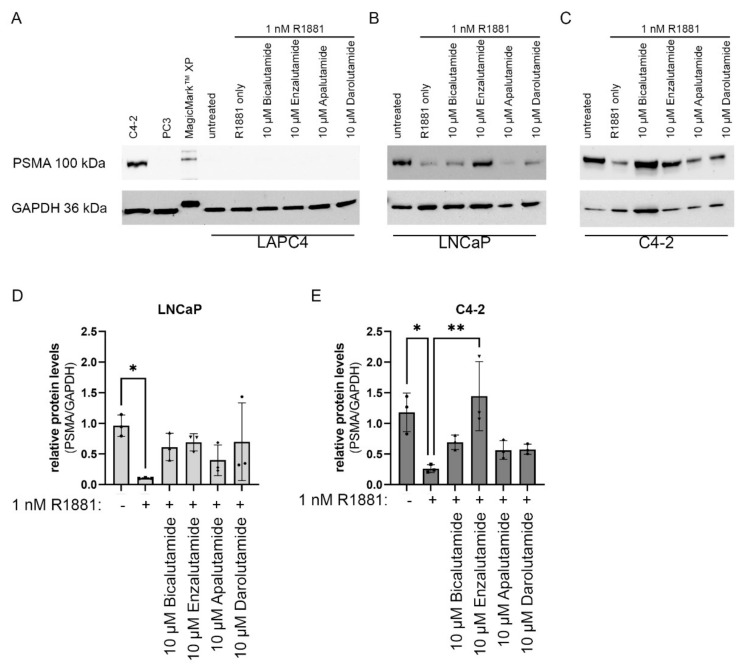
Impact of AR activity on PSMA protein levels**:** (**A**–**C**) Representative western blots of PSMA protein and GAPDH protein in the cell lines LAPC4 (A), PC3 (**A**), LNCaP (**B**), and C4-2 (**C**) after 1 nM R1881 and 10 µM bicalutamide, 10 µM enzalutamide, 10 µM apalutamide, or 10 µM darolutamide treatment. (**D**,**E**) Densiometric analysis of PSMA western blot analysis normalized to GAPDH after treatment with 1 nM R1881 and 10 µM bicalutamide, 10 µM enzalutamide, 10 µM apalutamide, or 10 µM darolutamide in the cell lines LNCaP (**D**) and C4-2 (**E**). Values are expressed as mean ± SD. All differences highlighted by asterisks were statistically significant (* *p* ≤ 0.05; ** *p* ≤ 0.01).

**Figure 6 ijms-23-01046-f006:**
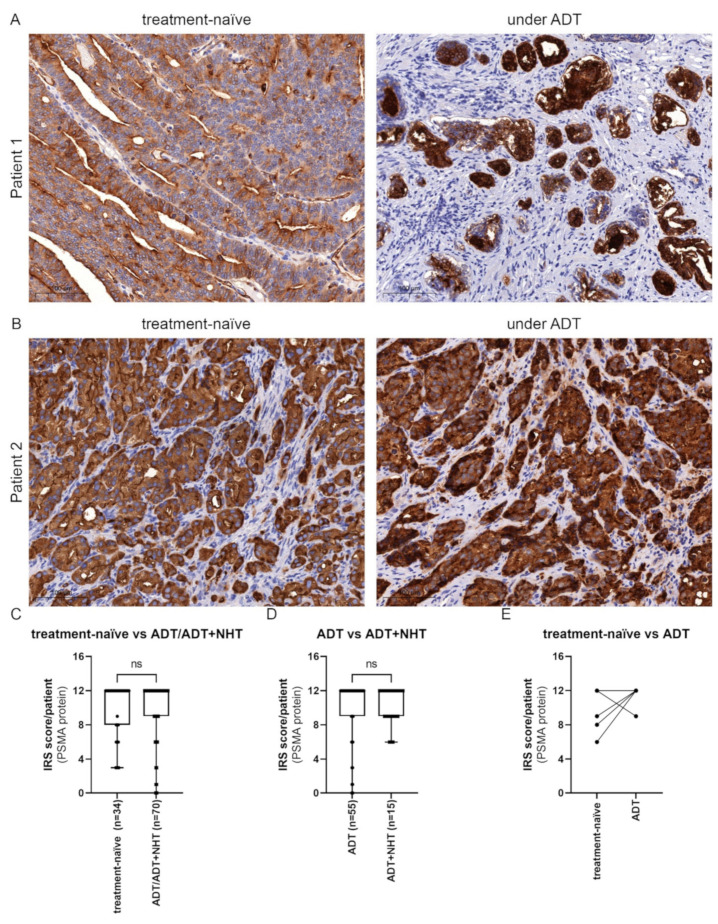
Modulation of AR activity does not generally change PSMA protein levels in PCa tissue: (**A**,**B**) Representative immunohistochemical staining of PSMA protein in treatment-naive and ADT treated PCa specimens of two patients. Scale bar represents 100 µm. (**C**) Quantification of PSMA within the HSPC and ADT/ADT+NHT cohort. (**D**) Quantification of PSMA within the ADT and ADT+NHT cohort. (**E**) Quantification of PSMA within the tissue of 6 patients before (treatment-naive) and under ADT. Values are expressed as a box whisker plot (min to max). ns: not significant.

**Table 1 ijms-23-01046-t001:** Characteristics of the used cell lines.

Name	AR Status	Origin Tissue	Reference
LNCaP	AR T877A, Androgen dependent	Lymph node	[19]
C4-2	AR T877A, ARv7 postive Androgen independent, LNCaP sub-cell line	Lymph node	[24]
LAPC4	AR wt, Androgen dependent	Lymph node	[27]
PC3	Cells express no AR protein, AR negative, small cell neuroendocrine carcinoma	Bone metastasis	[29,32]

**Table 2 ijms-23-01046-t002:** Antibodies used in the study.

Name	Company	Lot	Dilution
Monoclonal Mouse Anti-Human PSM IgG2b (Y-PSMA2)	Santa Cruz Biotechnology	I1611	1:10,000
Androgen Receptor (D6F11) XP Rabbit mAb	Cell Signaling Technology	9	1:5000
Mouse Monoclonal anti-GAPDH (6C5)	Novus Biologicals	19/05-G4cc-C5cc	1:10,000
Polyclonal Rabbit Anti-Mouse Immunoglobulins/HRP	Agilent Technologies	20066043	1:10,000
Polyclonal Swine Anti-Rabbit Immunoglobulins/HRP	Agilent Technologies	41289300	1:10,000

## Data Availability

Not applicable.

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
