# Peer review of "Impact of Androgen Receptor Activity on Prostate-Specific Membrane Antigen Expression in Prostate Cancer Cells"

_ijms, 2022, doi:10.3390/ijms23031046_

Round 1

Reviewer 1 Report

The manuscript entitled “Impact of Androgen Receptor-Mediated Gene Transactivation on Prostate-Specific Membrane Antigen Expression in Prostate Cancer Cells” by Sommer et al. was investigating the influence of androgen manipulation by R1881, Bicalutamide, Enzalutamide, Apalutamide, or Darolutamide on PSMA levels in vitro and comparing these findings with PSMA expression in patients. This is an interesting topic. However, the manuscript needs a minor revision before publication.

Cite Barfeld et al EBiomedicine 2017

FOLH1 is suppressed by R1881 (see Supplementary Table 4 in Barfeld et al EBiomedicine 2017). PSMA is directly suppressed by AR.

RESULTS

Explain what you mean by:

“modification” “of AR-mediated gene transactivation”

“new hormone therapy”?

Introduce BPH in line 189.

Figure 1. You have also measured FOLH1 mRNA and PSMA protein levels in the AR-negative PCa cell line PC3, add this info to the first sentence.

Lines 186-187: “Our results revealed a change in PSMA protein expression after modulation of the AR-mediated gene transactivation.”

Which data? Which change? In which patients? After which “modulation?

Author Response

On behalf of all authors, we would like to take this opportunity to express our sincere gratitude to the reviewers who identified areas of our manuscript that needed correction or modification. Their insightful comments have led to an improvement in our manuscript. Below you find the detailed response to the reviewers’ comments:

Cite Barfeld et al EBiomedicine 2017

FOLH1 is suppressed by R1881 (see Supplementary Table 4 in Barfeld et al EBiomedicine 2017). PSMA is directly suppressed by AR.

The information has been added to the manuscript, and the publication has been cited.

RESULTS

Explain what you mean by:

“modification” “of AR-mediated gene transactivation”

“Modification” was exchanged for “activation and inhibition” to describe the work step more precisely.

“new hormone therapy”?

“New hormone therapy” was changed to “novel hormonal therapy” and includes Second-Generation Antiandrogens and abiraterone. We included a definition in the manuscript.

Introduce BPH in line 189.

The abbreviation is now introduced in the manuscript.

Figure 1. You have also measured FOLH1 mRNA and PSMA protein levels in the AR-negative PCa cell line PC3, add this info to the first sentence.

We added the missing information

Lines 186-187: “Our results revealed a change in PSMA protein expression after modulation of the AR-mediated gene transactivation.”

Which data? Which change? In which patients? After which “modulation?

We changed the sentence into “Our western blot results (Figure 5) revealed a change in PSMA protein expression after activation and inhibition of the AR in LNCaP and C4-2”. Therefore all missing information should be available now.

Reviewer 2 Report

Abstract should be improved. Its understanding is very difficult. In background of abstract section please specify that FOLH1 represents the name of PSMA gene.

In introduction  section authors should describe better the features of prostate cancers and the therapeutic strategies available. It is well known that often PC progresses towards CRPC. I strongly suggest to introduce and discuss the current available therapeutic strategies and recent advancement in therapy. To this purpose, I highlight an interesting manuscript  focused on this relevant topic (DOI: 10.1016/j.critrevonc.2020.102992).

A brief description emphasizing the differences among the cell lines used is needed also in results section. LAPC4, LNCaP and C4-2 are AR positive. However, L4PC4 cells show similar PSMA protein expression level to PC3, but show differences in mRNA expression levels. How explain this behavior? What are the peculiar differences among the cell lines used?

 In the experiment setting reported in figure 2 D,E, F authors should indicate also R1881 alone.

Method describing AR mediated gene transactivation is lacking. Did you performed a Luc assay? Did you referred to a modulation of gene expression?In this case, the title is not pertinent and should be changed.

What are the differences among the drugs used? How do you explain the difference in their efficacy on FOLH1 expression levels?

Androgens display a dual effect through genomic and non-genomic pathways. Is it possible that the non- genomic pathway affect PSMA?

The paragraph “The AR is not mandatory for FOLH1 gene expression” seems in contrast with the aim of the manuscript.

Are the AR binding sites in FOLH1 are  canonical AR binding sequences?

What are the characteristics of  GSE11428 and GSE13332?

The paragraph “impact of AR …… protein levels” should be reported previously.

“transactivation” is used improperly.

In figure 5 indicate R1881 alone

Author Response

On behalf of all authors, we would like to take this opportunity to express our sincere gratitude to the reviewers who identified areas of our manuscript that needed correction or modification. Their insightful comments have led to an improvement in our manuscript. Below you find the detailed response to the reviewers’ comments:

Abstract should be improved. Its understanding is very difficult. In background of abstract section please specify that FOLH1 represents the name of PSMA gene.

The abstract was modified to improve comprehensibility.

In introduction  section authors should describe better the features of prostate cancers and the therapeutic strategies available. It is well known that often PC progresses towards CRPC. I strongly suggest to introduce and discuss the current available therapeutic strategies and recent advancement in therapy. To this purpose, I highlight an interesting manuscript  focused on this relevant topic (DOI: 10.1016/j.critrevonc.2020.102992).

A paragraph about prostate cancer and therapy options has been added according to the EAU guidelines.

A brief description emphasizing the differences among the cell lines used is needed also in results section. LAPC4, LNCaP and C4-2 are AR positive. However, L4PC4 cells show similar PSMA protein expression level to PC3, but show differences in mRNA expression levels. How explain this behavior? What are the peculiar differences among the cell lines used?

Information about the cell lines has been added to the introduction. Also, a justification for the selection was added in the Material and Method section.

 In the experiment setting reported in figure 2 D,E, F authors should indicate also R1881 alone.

The data already included the R1881 treatment. However, on closer inspection, the labelling was very confusing. Therefore, we have changed the labelling of the figure and hope that it is now more accurate and more precise.

Method describing AR mediated gene transactivation is lacking. Did you performed a Luc assay? Did you referred to a modulation of gene expression?In this case, the title is not pertinent and should be changed.

We agree with the reviewer that AR-mediated gene transactivation was an overinterpretation as we cannot prove that only the AR-mediated gene transactivation changes after treatment. Therefore, we changed the phrase into AR activation and defined the term in the introduction section. Also, we changed the title of the manuscript.

What are the differences among the drugs used? How do you explain the difference in their efficacy on FOLH1 expression levels?

We added information about the drugs into the Material and Method section. Cell line-specific effects of antiandrogens have been identified in a previous study and explained this with the AR mutation status and AR protein stability after antiandrogen treatment (PMID: 34575023). We added specific details to the discussion section of the manuscript.

Androgens display a dual effect through genomic and non-genomic pathways. Is it possible that the non- genomic pathway affect PSMA?

A non-genomic effect on PSMA is possible. Inhibition of the VEGF receptor tyrosine kinases by cediranib has also been reported to regulate FOLH1 in the Alveolar soft part sarcoma. As AR also regulates several kinases such as SRC and PI3K, there is also the possibility that the observed effects on FOLH1 mRNA are due to the non-genomic AR pathway. Therefore, we added the information into the manuscript.

The paragraph “The AR is not mandatory for FOLH1 gene expression” seems in contrast with the aim of the manuscript.

The paragraph was one of the results we discovered while investigating our aim. We will not change our study aim or hypothesis according to an unexpected result.

Are the AR binding sites in FOLH1 are  canonical AR binding sequences?

No complete canonical AR binding sequences have been identified in the gene. Therefore, we added the data to supplementary figure 1.

What are the characteristics of  GSE11428 and GSE13332?

We added the characteristics of the datasets to the material and method section.

The paragraph “impact of AR …… protein levels” should be reported previously.

We discussed this issue, and, in our opinion, we would like to keep the mRNA and ChIP Seq data together and the protein data. Therefore, the order of the data was not changed.

“transactivation” is used improperly.

We agreed and modified the manuscript as described earlier.

In figure 5 indicate R1881 alone

In line with figure 2, the R1881 alone data is already included. We have changed the labelling of the figure and hope that it is now more accurate and more precise.

Round 2

Reviewer 2 Report

Authors improved the manuscript according to my suggestions.